

# Automatic Georeferencing of Astronaut Auroral Photography

Maik Riechert[1,2], Andrew P. Walsh[3], Alexander Gerst[4], and Matthew G. G. T. Taylor[1]

[1]European Space Agency, European Space Technology Centre, Keplerlaan 1,2201AZ Noordwijk ZH,The Netherlands
[2]Now at Department of Meteorology, University of Reading, Reading, UK
[3]European Space Agency, European Space Astronomy Centre, PO Box 78, 28691 Villanueva de la Cañada, Madrid, Spain
[4]European Space Agency, European Astronaut Centre, Linder Hoehe, D-51147 Cologne, Germany

*Correspondence to:* Andrew Walsh (Andrew.Walsh@esa.int)

**Abstract.** Astronauts on board the International Space Station have taken thousands of high resolution, colour photographs of the aurora which could be made useful for research if their pointing information could be reconstructed. We describe a method to do this using the starfield in the images, and how the reconstructed pointing can then be used to georeference the images to a similar level of accuracy as existing all sky camera images. We have used this method to make available georeferenced auroral
images taken from the ISS, and here describe the resulting dataset, processing software, and how to access them.

## 1 Introduction

The aurora borealis and australis are spectacular displays of light concentrated in ovals around the north and south geomagnetic poles. Aurorae exhibit both slower, large spatial scale, variations that are driven in part by the interaction between the Earth's ionosphere, magnetosphere and the interplanetary magnetic field (e.g. Dungey, 1961; Akasofu, 1964), and faster, smaller scale
variations that are thought to be related to small scale physical processes, for example plasma waves or instabilities (e.g. Nishimura et al., 2010a). Thus, through studying the aurora it is possible to gain an insight into both the large scale behaviour of the coupled solar wind-magnetosphere-ionosphere system and the smaller scale plasmaphysical processes that operate there, and indeed the interaction between the two.

Investigation of the aurorae has many facets, involving the use of ground-based ionospheric radar and magnetometers (e.g.
Chisham et al., 2007; Mann et al., 2008; McCrea et al., 2015), single (e.g. Paschmann et al., 2002) and multi-point (e.g. Forsyth and Fazakerley, 2013, and references therein) space-based in situ measurements of charged particles and electromagnetic fields, and of course through direct imaging of the aurora from both space and the ground. Space-based images of the aurora are comparatively rare, with few operating missions providing auroral imagery. In fact since the end of the IMAGE (Burch, 2000) and Polar (Acuña et al., 1995) missions in 2005 and 2008, there have been no dedicated global auroral imagers, although smaller
scale images have been available from Reimei (Obuchi et al., 2008), for example. Networks of ground-based auroral observatories (e.g. Mende et al., 2008; Syrjäsuo et al., 1998) can provide both large and small scale auroral observations, without the telemetry constraints that can limit space-based auroral imaging. They do have the disadvantage however of sometimes being obscured by clouds or snow, and necessarily have limited coverage in the southern hemisphere.



Astronauts on board the International Space Station (ISS) have taken thousands of images of the aurora borealis and aurora australis (fig. 1), often in higher resolution than is available from other sources, and collected into sequences encompassing a single passage of the ISS over the auroral oval. The orbit of the ISS has an inclination of 51.5 degrees from the geographic equator, meaning that the space station often skims the auroral oval in a roughly westerly direction, in contrast to the DMSP

spacecraft, for example, whose more highly inclined orbits mean that they cut through the oval perpendicular to it. This is illustrated in fig. 2. The orbital velocity of the ISS ($\sim 7.5\,\mathrm{km\,s^{-1}}$) allows it to pass over a significant proportion of the auroral oval in $\sim 5$ minutes, thus a single sequence has the potential to give both a global scale snapshot of auroral morphology, and details of smaller scale auroral forms and dynamics. Furthermore, because of the comparative lack of ground based auroral imagers in the southern hemisphere, ISS auroral photographs can provide valuable data on the aurora australis.

While the ISS has a lot of advantageous characteristics for an auroral observatory, to date the images that have been taken were not intended to be used in research. They have been taken manually by astronauts, using commercial Digital Single Lens Reflector (DSLR) cameras that do not have fixed pointing, or indeed pointing information available. Here, we present a method by which photographs of the aurora taken by astronauts on board the International Space Station can be georeferenced (i.e. have their pointing reconstructed) and processed, using only free and open source software and data, to make them useful for

research. We also describe the resulting dataset that has been made available, and how to access it. Details of the open source software libraries and data used are listed in the acknowledgements.

## 2   Astronaut Photography of the Aurorae

On the ISS, photography of aurorae is typically acquired in the crew's free time, as it is not a time-lined activity coordinated with the Mission Control Centre. Imagery is typically obtained for aesthetic reasons. Onboard conditions, due to the activity

not being part of a controlled experiment, do not allow to control or measure image parameters such as pointing or exact camera location within the ISS stack of modules.

The most typical setup is to mount one of the common ISS crew cameras on a support arm ("Bogen arm") in the ISS' Cupola window module. Until May 2015, one of the ISS modules ("PMM" - the permanent multipurpose module) was berthed to the Node 1 nadir position, partially blocking the view to the starboard side of the vehicle, which corresponds to a general Southern

direction, since ISS is typically steered in an LVLH (local velocity local horizon) attitude. Thus, images of Northern Aurorae were typically easier to obtain. As a secondary effect, ISS crew working hours correspond to UTC daylight working hours, allowing for a more convenient observation time of Southern Aurorae, as cameras can be set up in small breaks between time line activities.

### 2.1   ISS Camera Setup

For aurora photo sequences obtained by the US operated segment crew, Nikon digital SLR cameras are used (D2XS, D3S, D4S). Cameras are typically used on board for several months to a few years, until pixel damage caused by the elevated radiation environment strongly impacts image quality.

Sequences are typically set up with a timer interval between 1 to 3 seconds, limited by the camera's technical capability, especially at exposure times in the order of 0.5 seconds. The cameras feature an integrated dark field sensor correction, which was sometimes used at the cost of a lower frame rate.

For the newest available camera model, the Nikon D4S, typical settings are ISO 5600 - 12800, exposure times between 0.1 and 1 seconds, and a fully open aperture limited by the lens speed. Typical lenses included 28 mm f1.4 and 17-35 mm f2.8.

Lenses are focused to infinity, which was typically done during the previous day pass on a part of Earth structure, or during a night pass by manually focusing on a star using the camera's live mode display in its maximum zoom setting.

The optical quality of Cupola windows is generally high, although it is strongly affected by the relatively poor optical quality of the window scratch protection panes, which fully cover the windows, and which could until recently not be removed for photography.

Synchronising the onboard cameras' built-in clocks to UTC is typically not an operational requirement, and due to the time-consuming process given the large number of available cameras, it is only performed once every few weeks. Natural drift of camera clocks thus leads to typical time errors in the order of several seconds to several tens of seconds.

Data on the camera flash cards are typically downloaded via the ISS Ku Band link within a few days of obtaining the images.

## 2.2   Image Availability

Astronaut photography of the Earth, including that of the aurora is made available by the NASA Johnson Space Center Earth Science and Remote Sensing unit. All images have a unique ID based on mission, expedition camera roll and frame number and are searchable by this frame number. Some images are also catalogued based on the features (e.g. a particular city or geographic location) in the image although this isn't necessarily true for all auroral images. Typically, we process image sequences of the aurora highlighted by the Earth Science and Remote Sensing Unit that are placed on a specific web page. Images are typically available in JPG and RAW format. An advantage of using the RAW images rather than the JPGs is that the JPGs are optimised frame by frame so are not always consistently processed across a sequence, making comparisons difficult. Another advantage of using the RAW images is that dead and hot pixels can be removed more easily from the images (see below).

## 3   Automatic Georeferencing Technique

In order to make the astronaut photographs useful for research, their pointing must be reconstructed and the images must then be projected onto a lat/lon grid, assuming a fixed emission height for the aurora. Pointing reconstruction is accomplished through automatically identifying the starfield in each image using the Astrometry.net software (Lang et al., 2010). However, the software is not designed for use on images that contain sections of the Earth, the aurora and indeed parts of the structure of the ISS, so some preprocessing is needed before the images can be successfully georeferenced. These preprocessing steps are described below.



### 3.1 Image Pre-Processing

#### 3.1.1 Lens Distortion Correction

The RAW images are not usually corrected for lens distortion, and the manufacturers of the lenses used do not always make their distortion profiles publicly available. This presents a problem since astrometry of a distorted image is less accurate than

of a corrected image. Some lenses have community-provided distortion profiles available[1], and we use these to correct the lens distortion in the images wherever possible. Where no community-provided distortion profiles exist, or if the profiles are not sufficiently accurate, we use the ISS image sequences themselves to produce an adequate distortion profile from scratch. This is done through comparing the positions of manually identified fixed points (e.g. stars) in overlapping images and is a technique often used in automatically constructing image panoramas (e.g Stein, 1997; Hugemann, 2010). While quantitative estimates

of the accuracy of the distortion corrections are difficult without properly calibrated images (And hence prior knowledge of the true distortion profile) for comparison, a usable profile should produce astrometric solutions where the accuracy of the star locations is uniform across the image. Note that additional small distortion from the windows of the ISS cannot easily be quantified since the position of the camera is not fixed, meaning solutions will necessarily be imperfect.

#### 3.1.2 Hot/Dead Pixel Repair

The goals of bad (i.e. hot/stuck or dead) pixel detection and repair are to prevent Astrometry.net from extracting false stars and avoid invalid values in the scientific image areas, that is, the earth/aurora. Camera sensors on the ISS deteriorate quite quickly as the cameras are not radiation hard and are used nearly every day, which means that the shutters are often open and the increased radiation then damages the sensors. Between ∼10,000 to ∼300,000 bad pixels were observed in typical 12 MP aurora images.

Hot pixels are those which are stuck at an often high value, such that the resulting interpolated pixels appear green, red, or blue. It can happen that hot pixels get unstuck.

    Dead pixels are permanently damaged and will be at zero value. The resulting interpolated pixels appear in different colours or are just darker.

    We detect bad pixels by finding outliers compared to a 3x3 pixel window median filtered image for each RAW channel, that

is, based on the uninterpolated Bayer RGBG matrix. The idea is that directly neighbouring pixels of the same colour channel should be in the same value range. An outlier may be caused by a bad pixel or by very high contrast, high noise, or other artifacts. To differentiate bad pixels from other effects, we analyse multiple images from the same sequence and only those bad pixel candidates which appear in over 90% of all images are considered as confirmed. For this to work reliably the image contents should be different across images, otherwise a non-moving structure may be falsely detected.

The detected hot and dead pixels in each image channel are corrected by replacing the values with the equivalent value from the 3x3 median filtered image channel.

---

[1]http://lensfun.sourceforge.net/



For our purposes, the important parts of the image are the earth/aurora and the starfield. Both are moving and are therefore perfect for detecting bad pixels. Using this method, non-moving spacecraft structures may be misdetected as bad pixels, but fortunately this is not critical in any way.

Note that the ISS JPEGs available from the Earth Observation Laboratory often have bad pixels corrected already. If not, then it is also not possible to do so as the detection relies on the uninterpolated RAW Bayer pattern pixels. After interpolation, in the JPEG images, bad pixels turn into small crosses or other artefacts.

### 3.1.3 Image Masking

In order for astrometry.net to successfully identify the stars in an image, it needs to estimate the background level and also not extract too many false stars from the image. For the ISS images, this means that the areas of the image containing the Earth and aurora, and the structure of the ISS must be masked so only the starfield remains. The image itself must be kept at its native dimensions so scale will be correctly returned.

The simplest way of masking an image is to manually select an area of the image that contains only starfield, and use that as input to astrometry.net. This has the advantage that it allows control over which part is used for star extraction. The disadvantage is that it requires manual input and is hence time consuming.

We have also developed an automatic masking procedure that uses a combination of image processing and recognition techniques that try to automatically find the starfield in the image while excluding other parts like the earth, aurora, and spacecraft structures. This method is block-based and may lead to bad results when only a small proportion of the image contains starfield, e.g. the upper 10% of the image. The following outlines how the method works:

1. A histogram of the image is calculated and the histogram values are smoothed using a simple unweighted sliding-average over three neighbours. The histogram covers each brightness level of the image, from 0 to 255.

2. The value at the top of the first peak in the smoothed histogram is considered as the average brightness level of the starfield background (i.e. all the dark areas around the stars). This assumes that the starfield background is the darkest part of the image. The maximum value of the peak, $b$, is identified by taking the first value at which the gradient of the histogram is negative (The histogram has been smoothed in order to avoid noise at low brightness levels).

3. The image is converted to black and white by thresholding with a constant value $t = b + f$, where $f$ is a variable offset (initially 20), up to a maximum value of $t = 150$.

4. The areas of the image containing starfield are then roughly identified using contours of this binary image (see section 3.1.4) and step 3 is repeated with $f$ increasing by 20 in each iteration until the percentage of assumed starfield is above 10% of the image or $f > 100$. The iteration is necessary for cases when the first histogram spike did not correspond to the actual starfield background but e.g. to some very dark part of the earth. This method obviously is not foolproof but seems to work for many situations.





5. The output of step 4 is then refined through further thresholding and line detection to remove e.g. dark areas of spacecraft structure (see section 3.1.5), producing the final masked image.

Figure 3 gives examples of the various stages of the masking process using the same image as fig. 1. Figure 3A is the black and white output of step 3. Figure 3B is the contour calculated in step 3, applied to the original image. Figure 3C shows some spacecraft structure detected as described in step 5 (the red lines), that have then been removed in fig. 3D, which has been brightened for clarity.

### 3.1.4 Contour Detection of Starfield Areas

Extreme outer contours are extracted from the binary image. All contours are classified as either "big", "small and long", or "small and short". A contour is considered "big" if its area is greater than 0.0013% of the whole image size. A contour is considered "long" if its area is greater than 20 pixels and the long side of the minimum area rotated rectangle covering the contour is at least 5 times as long as the short side. The idea of this classification scheme is to identify everything that is most likely not part of the starfield. Big structures like earth or spacecraft structures should be identified as "big" contours, and the remaining "small" objects are either tiny long spacecraft structures or stars. As star contours are usually not long, the combination of "small" and "long" is used to identify tiny long artificial structures. Note that faint stars are often not detected at all as a contour, therefore contours classified as "small and short" are not used for any further analysis but merely for separating against the "small and long" contours.

The image is then conceptually split up into 16x12 (if the image width is divisible by 16 and the height by 12) or alternatively 16x8 blocks. This results in roughly square blocks for the typical DSLR camera image formats.

If the top end of the contour with the biggest area is in the bottom two thirds and the lower end within the bottom half of the image, then all blocks below the top end are removed from the assumed starfield. This contour is assumed to be (part of) the illuminated earth/aurora. If the conditions do not apply, then all blocks below the bottom half of the image are removed from the assumed starfield as a generic fall-back. This step tries to remove as much of the earth as possible, assuming that the earth is in the lower part of the image and that the image is more or less horizontally oriented to the horizon. Finally, any other blocks which contain "big" or "small and long" contours are also removed from the assumed starfield.

### 3.1.5 Refining the Mask

After broadly masking the starfield, there are several additional steps that try to remove areas of the assumed starfield which are in fact not starfield but other structures such as earth or spacecraft structures.

The contour-masked input image (not the black and white image used from calculating the contours) is adaptively thresholded with a 89x89 pixel moving window. The resulting binary image greatly emphasises irregular and previously very dark structures. The structures are amplified further by running a median filter with window=3 on the binary image. In this binary image, line segments are searched for using a probabilistic Hough transform (Kiryati et al., 1991) with parameters $\rho = 1$, $\theta = \pi/180$, $threshold = 200$, $minLineLength = 100$, $maxLineGap = 4$. Every image block which contains (part of) a de-



tected line is removed from the assumed starfield. This method works well to detect dark spacecraft structures (e.g. solar panels which are brightly illuminated only in some periods of the sequence).

An averaging filter with a 3x3 pixel moving window is then run on the broadly masked input image. Every image block which does not contain pixel values above either 30 or $b + 20$, whichever is higher, is removed from the assumed starfield. This step further tries to remove dark non-starfield areas (e.g. unlit parts of the Earth) which were previously detected. A potential problem with this step is that it removes actual starfield blocks that contain only very faint (dark) stars, which likely still would have been extracted by astrometry.net. This is illustrated in fig. 3D, where the red squares show blocks containing either spacecraft structure or dark starfield that would be removed in this step.

Every remaining starfield block which is surrounded only by non-starfield blocks is removed from the assumed starfield. The assumption is that starfield blocks are typically adjacent to at least one other starfield block. A single assumed starfield block which is only surrounded by non-starfield blocks is most likely not actual starfield but rather some dark area of the earth. A potential problem of this step is that it may remove actual starfield blocks in cases when only a very small percentage of the image is starfield.

## 3.2 Astrometry

Each masked image is then passed to Astrometry.net in order to identify the stars in the image, and hence its orientation and plate scale. An astrometric solution following the WCS (World Coordinate System) conventions (e.g. Greisen and Calabretta, 2002) is defined by the right ascension and declination of the image centre together with the celestial pixel scale and rotation angle. The image shear and distortion can also be defined as part of the astrometric solution, but in our case the images do not have shear and the distortion has been corrected before-hand.

Astrometry.net is fully described by Lang et al. (2010), but briefly, it works by extracting stars from an image in groups of four ("quads"). A geometric hash or each of these quads is then calculated and compared with catalogues containing precalculated hashes of many different quads of stars. The Astrometry.net software has two sets of star catalogues which can be used for finding an astrometric solution, 2MASS (Skrutskie et al., 2006) and Tycho-2 (Høg et al., 2000). The Tycho-2 catalog (here the 4100 series) is recommended by the Astrometry.net authors for widefield images, which the ISS images are. Thus, the Tycho-2 catalog is used for this project. Only the large-scale index files 4111 to 4119 (maximum) are used, as smaller scales (down to 4107) are not needed. A completely blind search for a match in the reference catalogue (Tycho-2) is fast when enough bright stars are available but can also be very slow. We narrow down the search space by giving Astrometry.net a range of possible celestial pixel scales. For this, the focal length $f$ corresponding to 35mm film is read from the EXIF header of the image and the pixel scale $s$ is calculated as $s = (35/w)/(f \times r)$ where $w$ is the image width in pixels and $r$ is 0.9 for calculating the lower bound and 1.1 for the upper bound. The resulting radian values are converted to arcseconds and passed to astrometry.net. The upper and lower bounds were empirically chosen to give the highest chance of successful astrometry in a reasonable processing time.

When using Astrometry.net with default parameters it only works well for photographs taken with telescopes from the ground on earth. The software assumes that the sky has a smoothly varying background caused by the atmosphere, which is




automatically removed. Also, it assumes that the centre of each star is the centre of a point spread function with 1 pixel of error. These assumptions do not hold for ISS auroral photography as (a) there is no atmosphere and (b) there are startrails due to the speed of the ISS and the necessarily long exposure times (0.2-1s). To correct these errors we disable background substraction and increase the pixel error of the point spread function to 10.

5     Another issue is that Astrometry.net tries to estimate the noise level, $\sigma$, in the image. The algorithm used is confused by our masked images. The non-starfield areas are painted pure black, and this can sometimes lead to incorrect $\sigma$ estimation. Instead, we estimate $\sigma$ ourselves from the largest starfield rectangle in the masked image using the algorithm described by Immerkær (1996). The output of this algorithm is modified with $\sigma = max(0.9, 2.5\,\sigma)$ as it was often estimated too low.

    Example output from Astrometry.net is given in fig. 4. Here green circles denote reference stars and red circles mark stars 10 extracted from the image. Where these are offset from one another, lens or window distortion has not been completely removed. Unfortunately we have no way to correct any residual distortion, however by comparing the astrometric solution of several images (specifically the celestial sphere pixel scale) in a sequence one can identify whether or not any residual distortion is likely to cause large errors. See Section 4.1 for details.

    The Astrometry.net parameters that have been changed from default, and the values used, are described in Appendix A.

## 15   3.3   Mapping

Once the astrometric solution of an image is determined it can be georeferenced at an arbitrary altitude using the image timestamp and the corresponding ISS position, producing geodetic latitude and longitude coordinates for each pixel centre and corner, and elevation angles for each pixel centre. The elevation is the angle between the line of sight and the normal to the intersection point. It ranges from 90° (nadir) to 0° (horizon).

20     The following mentions only "pixel" but refers both to pixel corner and centre.

    First, the right ascension and declination in the ICRS (International Celestial Reference System) frame is calculated for each pixel. These are then transformed to Cartesian coordinates and represent the line-of-sight vector for each pixel. From the image timestamp, the ISS position is determined in J2000 Cartesian coordinates. The ISS position is taken as the assumed camera position. For each pixel the Cartesian intersection point with earth (if any) closest to the camera is calculated using the 25 line starting from the camera position and following the line-of-sight vector. The line of sight vectors are not transformed to J2000 from ICRS. The difference between ICRS and J2000 is ∼0.01 arcsec and is thus negligible when compared to the pixel resolution of the ISS images (20-100 arcsec/px).

    The intersection points are in J2000 coordinates and have to be transformed to the Earth-Centered Earth-Fixed (ECEF) system via simple spherical rotation using a rotation matrix that is determined by the timestamp of the image. The ECEF 30 coordinates are then transformed to geodetic latitude and longitude coordinates using Bowring's method (Bowring, 1985).

    For ground altitude, the WGS84 reference ellipsoid is used as a model of the earth. For a higher altitude $h$, the model of the earth is defined as an ellipsoid with an equatorial axis of $wgs84A + h$ and a polar axis of $wgs84B + h$. This ellipsoid is an approximation of a surface which is at a constant height $h$ above the WGS84 reference ellipsoid. The mapping result is nearly



identical to approximating the intersection surface as a sphere. However, it is still important for identifying geographic features in the mapped image.

To produce MLAT/MLT coordinates the intersection points in the J2000 system are rotated to the SM system, then converted to Solar Magnetic (SM) latitude and longitude, where SM latitude = MLat, and MLT = lon*(24/360)+12. The geomagnetic

north pole necessary for transforming to the SM system is calculated using the first three IGRF (Thébault et al., 2015) coefficients interpolated to the image time.

A stereographic projection of the example image shown in fig. 1, mapped to 110km altitude, is plotted in geographic coordinates in fig. 5. Note that to avoid projection effects at the edge of the effective field of view and in common with ground-based all sky imagers (e.g. Mende et al., 2008), we have not plotted elevations below 10°.

## 4 Sources of Error and Error Correction

The quality of a georeferencing is directly determined by the quality of the astrometry, the accuracy of the image timestamps, and the amount of window distortion. Here we describe methods by which the accuracy of the astrometry and image timestamps can be improved. Window distortion cannot be corrected without a dedicated calibration campaign on board the ISS, which is beyond the scope of this project.

### 4.1 Errors and Instabilities in the Astrometric Solution

The accuracy of the astrometry for an individual frame can be qualitatively assessed by examining plots similar to fig. 4. When the green circles (reference stars) rarely match with the red circles (extracted stars) then this either means that the lens distortion profile is not accurate or the window distortion is noticeable.

When the right ascension, declination and rotation values vary a lot over an image sequence this can also point to a problem
with distortion. Small variations are normal and can occur for several reasons. Astrometry.net looks for a matching triangle of stars using geometric hashing (Lang et al., 2010). This initial solution is then tuned to better match other extracted stars. The tuning may fail if too many, too few, or false stars were extracted. Due to the star trails in ISS images, the tuning also doesn't have the best starting conditions.

The standard deviation of the celestial pixel scale of the astrometric solutions of a sequence usually gives a good idea about
the stability of the solutions. Stable solutions over a sequence in turn imply that the lens distortion profile is accurate. This is because the pixel scale of an image is determined from the size of the matched star triangle. Over an image sequence as the star triangle moves across the field of view, the size of the triangle and therefore calculated pixel scale for each image will vary based on the local distortion. Conversely, when the image has no distortion, then the size of the triangle and hence pixel scale will be the same in each area. After some experimentation, it was found that a standard deviation of less than 0.15 arcsec/px is
very good, less than 0.1 arcsec/px excellent. Anything above 0.5 arcsec/px is very bad and is a typical range when distortion was not corrected.



As well as small variations in pixel scale from any remaining distortion, sometimes Astrometry.net finds a false solution, which may be due to the relaxed pixel error parameter we use (10 instead of 1px). These outliers are easy to see as spikes in the time series of pixel scale, for example, and are not considered when assessing the standard deviation.

As well as the pixel scale, other astrometric parameters over a whole sequence are typically slightly noisy. This is caused by many factors, e.g. the star trails, a slight window distortion, or a failed tuning attempt by Astrometry.net. When it is clear from the original sequence movie that the camera was did not move during the sequence, this noise can be removed through a polynomial fit to each of the astrometric values (right ascension, declination, rotation), and also by fixing the celestial pixel scale to a single value, i.e. its median. We assume each astrometric parameter is represented by a polynomial of degree 3, which is a function of elapsed time (in seconds) from the timestamp of the first image of the sequence. For a polynomial fit to be considered successful the root mean square error of the fit for each of the right ascension, declination and rotation should be below 0.05deg.

Example astrometric parameters, and the fits to them are shown in fig. 6. Figure 6A is the celestial sphere pixel scale (note the outliers early in the sequence); fig. 6B the right ascension; 6C the declination and 6D the rotation. In the latter three cases the red trace is the polynomial fit and the black trace the parameters as determined by Astrometry.net.

Note that polynomial fits are only attempted when the astrometric solutions are already of high quality, meaning that the pixel scale standard deviation is low and the distortion is very well corrected. Otherwise, a polynomial fit could worsen existing solutions. Another advantage of using the polynomial fits is that accurate solutions for unsolved frames (and outliers) are determined implicitly.

## 4.2 Timestamp Accuracy

The internal clocks of the cameras used on the ISS are not automatically synchronised to UTC. Instead they are set manually by the astronauts and can be inaccurate by up to around 1 minute due to clock drift. An incorrect timestamp results in inaccurate georeferencing since the ISS position used as an origin point will be incorrect. As such, timestamps have to be corrected manually once per image sequence. This can be accomplished by comparing a georeferenced image with known reference points and adjusting the timestamps until features in the image match the reference points. We have identified two potential ways of doing this for the ISS images:

The correction of the timestamps is most accurate when city lights are visible somewhere in the image sequence. By georeferencing an image to ground altitude with a given shifted timestamp and visualising it on a geographical map, the city lights can be aligned to reference city markers that are overlaid on the image. This typically allows to correct the timestamps to within 1s or less. In most cases the city lights cannot be perfectly aligned and have some residual offset. The current assumption is that this error is caused by window distortion. The effect of a 13 second shift in timestamp can be seen in fig. 7, where the left hand plot has the timestamps from the image EXIF data and the right hand plot has been shifted to align the city lights as well as possible with known city locations.

If no city lights are visible (or as an additional check) then the Earth's horizon can be used as a reference, by overplotting the expected horizon location on the original image. This is not always possible, since the horizon is not always clearly visible.



Sometimes it is the case that the horizon appears to be identifiable in the image but is in fact covered by smooth cloud (typically about 8-14km high). The accuracy of this method also depends on the angle between the camera look-direction and the ISS velocity. If the camera is pointing parallel to the ISS velocity then an incorrect timestamp will result in the horizon being shifted lower or higher. This allows for more accurate correction than if the camera was pointing closer to perpendicular to the

direction of motion, in which case the horizon will appear lifted on one side and lowered on the other – in this case because of projection effects the correction will not be as accurate. In general, using the horizon allows to correct the timestamps to within around 2-6s.

Special care has to be taken when correcting the horizon at sunrise. The brighter the horizon gets the higher the error is as the brightness "overflows" over the real horizon. In this case it is best to use multiple frames where the sun is illuminating the

horizon just enough in different parts, e.g. three frames where the left, right, and middle part of the horizon can be clearly seen.

An example of timestamp correction using the horizon is given in figs. 8 and 9, where the horizon is drawn in white over the image. Figure 8 has uncorrected timestamps. Figure 9 has timestamps shifted by 25 seconds.

If both city lights and the horizon are visible, then in ideal conditions the correction using the horizon and the city lights should match each other. In some sequences this is not the case and could be due to the window distortion.

## 15  4.3  Emission altitude and projection effects

It is also important to recall that, in common with the processing of other auroral images, this technique assumes a constant, single emission height for the aurora, which is not the case. This assumption causes the largest errors at low elevations and is minimised when nadir pointing. This is illustrated (right) and quantified (left) for the specific case of these ISS images in fig. 10, where the angular extent of a feature, along the projection of the line of sight at a fixed altitude, ($\Delta\theta$), is plotted as a function

of that feature's extent in altitude ($\Delta h$). The different traces denote the curves for different lines of sight ($\phi$; $0° =$ nadir). For example, an auroral arc with a vertical extent of $40\,km$ (assuming the lowest visible emission at $110\,km$), viewed at an angle of $60°$ from nadir will be projected over $0.4°$. Note that the same angular offset from 'true' position would be observed if the emission came instead from a narrow range of altitude centred on $150\,km$. In the illustration to the right of fig. 10, $r$ represents the altitude of the ISS and $h$ the lowest extent of the emission height. $\theta$ is the angular offset of the true emission location from

the sub-spacecraft point and $\Delta\phi$ the change in line-of-sight angle from the bottom to the top of the feature. Unfortunately, while we can estimate the degree of projection of a hypothetical auroral arc, correcting projection effects is not possible when only one image source is available. In the small number of cases where these images overlap with ground-based cameras, accurately calculating the altitude extent of auroral features may be possible, but it is beyond the scope of this paper.

## 4.4  Other errors

Even after applying the above corrections, the georeferencing will not be perfect. This is most obviously seen in the discrepancy between city light locations and city reference locations (fig. 7). This is likely due to a combination of uncorrectable distortion from the windows of the ISS, small errors in the ISS ephemeris and the fact that we use WGS84 ground altitude, rather than true altitude at the reference city locations. Another potential complication is that coastal squid fleet fishing lights are also easily



visible from low Earth orbit, and could be confused with city lights. That said, a comparison between our georeferencings and ground based auroral imagery (section 5) shows good agreement.

## 5    Comparison with Ground Based Images

During a pass of the ISS over North America on 4 February 2012 the aurora was visible and photographed from the ISS and

also visible in the THEMIS array of all sky imagers (Mende et al., 2008). This gives us an opportunity to test the accuracy of our georeferencing. A THEMIS mosaic, containing images from the GILL, SNKQ and KUUJ imagers, from west to east, is plotted in Fig. 11 in greyscale with a subfield of the green channel of a georeferenced ISS image (frame number ISS030-E-85117, acquired at the same time) overplotted in green. The selected subfield of the ISS image is entirely encompassed in the SNKQ field of view. The auroral oval can be seen to continue smoothly from the THEMIS imagery onto the ISS image,

suggesting that, at the large scale, our georeferencing is accurate. A smaller arc is also visible starting at the northern edge of the SNKQ field of view and continuing smoothly into the ISS image.

This is further examined in fig. 12 where the SNKQ image is plotted in panel A, a slightly larger subfield of the georeferenced ISS image in panel B and the original ISS Image in panel C. The red symbols in panel A were positioned to follow the small 'j' shaped arc and are repeated at precisely the same latitudes and longitudes in panel B. The equivalent arc in the original image

is denoted with a red arrow. The red-marked arc appears to be shifted in the ISS image by around 12 arcminutes (we believe that quoting an angular separation is appropriate here because of the uncertainty in emission height - an accurate linear distance is difficult calculate). This is likely to be because of an imperfect estimate of emission height, and also projection effects since the arc is close to the horizon, as explained in section 4.3 (see panel C). Conversely, the position of the northern edge of the main auroral oval (blue symbols in A and B) is in almost perfect agreement between the THEMIS ASI and ISS image. This

feature is much closer to ISS nadir and shows less vertical structure in the original ISS image (blue arrow in panel C).

A more extreme example of how projection effects can result in incorrect interpretation of auroral features in georeferenced images can be seen by examining the feature highlighted by the magenta arrow in each panel of fig. 12. Examining the original ISS image shows that the arc in fact has significant vertical extent and may even be an edge-on view of the red-marked arc. However, the feature appears to be a $\sim$ North West to South East aligned arc in the THEMIS ASI, and a $\sim$ North East to South

West aligned arc in the ISS image. Note that in both cases this is the direction approximately radially from zenith (ASI) or nadir (ISS), i.e. projected along the line of sight from the camera as one would expect from a vertical structure (see section 4.3). Thus care has to be taken when performing detailed analysis of these data and other projected images, especially when looking for North-South aligned arcs (e.g. Nishimura et al., 2010b).

## 6    Available Data Products

A sequence of $\sim 300$ images takes roughly six hours to plot, not including the time required to download the data. At the time of writing 4383 images, taken between 2011 and 2015, have been processed and intend to keep expanding the dataset as time




and resources allow. We have made the georeferenced image sequences available through http://cosmos.esa.int/arrrgh. Movies of the sequences, frame by frame plots (e.g. Fig. 5) in geographic and MLAT/MLT coordinates and summary 'scanline' plots which are constructed from a thin slice of each frame, taken at a constant position and orientation, and are useful to gain on overview of auroral activity during each pass of the ISS over the aurora. An example of the scanline plot is given in Fig. 13.

All of the available plots assume an emission height of 110 km for the aurora.

We have also produced an open source software toolkit, 'AUROMAT' (AUROra Mapping Toolkit), written in python, and a simple API that allows the community to download the images and pointing information. Thus it is possible for users to produce plots themselves that are projected to an arbitrary altitude, rather than the 110 km that we assume. The software toolkit is also capable of producing georeferenced images stored in CDF and netCDF files for comparison with other data. We do not

make the digital data available directly because a full-resolution image, the latitude and longitude of each pixel corner, and other necessary metadata results in a file size of $\sim$300 MB per frame, so generating the files locally as needed is much more efficient.

# 7 Conclusions

We have described a method of georeferencing astronaut auroral photography using only the starfield in each image, its EXIF

metadata and the location of the International Space Station when the image was taken. Our georeferenced images are of high resolution compared to much auroral imagery and the georeferencing can agree with projections of THEMIS ASI images to within 12 arcminutes or better, depending on how much vertical structure is visible in the original image and the proximity to the horizon of a given auroral form.

We have made available all of the images that have been processed to date at http://cosmos.esa.int/arrrgh along with the

necessary software for producing the images, and digital data. This entire project was accomplished using only free and open source software.

# Appendix A: Astrometry.net Input Parameters

We use the following input parameters to Astrometry.net in order to maximise the proportion of images that are solved: Background subtraction is switched off by using the –no-background-subtraction parameter. Using –pixel-error 10 helps take

into account short star trails caused by the long exposure times. Solving is sped up by limiting the range of pixel scales as follows: –scale-low 45.678 –scale-high 48.901 –scale-units arcsecperpix. We set our calculated noise level using –sigma.

The author of astrometry.net suggests to downsample the image by factor 2 using the –downsample 2 parameter in case the star extraction fails. The effect of downsampling is that the noise is reduced and the stars become smaller, which may be better for the used point spread function which seems to be in general too small for high resolutions. In our processing pipeline, we

try 2x, 4x and no downsampling. Although with 4x downsampling the accuracy is slightly reduced, it doesn't seem to have a noticeable effect. In most cases, 2x downsampling leads to a result and this setting is tried first.



Another important parameter is –crpix-center as it forces the centre of the tangent-plane projection (TAN) to be the image centre. While in general this cannot be assumed for arbitrary (space) telescopes, it is always true for "normal" cameras where the optical boresight is the centre of the image, and with that we "help" astrometry.net a bit. Otherwise it would have to determine the projection centre itself.

5 As the input images are distortion-corrected we disable astrometry.net's integrated distortion correction with –no-tweak. Note that the integrated correction would not work at all for the ISS images as you would need more stars over the whole image to automatically create a distortion profile.

*Acknowledgements.* Original images are courtesy of the Earth Science and Remote Sensing Unit, NASA Johnson Space Center (http:// eol.jsc.nasa.gov). Georeferencing was accomplished using astrometry.net software (http://www.astrometry.net), Lens distortion correction

10 used the lensfun database (http://lensfun.sourceforge.net) or Hugin (hugin.sourceforge.net). City locations are from Natural Earth (http: //www.naturalearthdata.com). We acknowledge the use of the following software libraries: CXFORM, numpy, spacepy, matplotlib, basemap, libraw, rawpy. MR was funded by the ESA YGT Programme (http://www.esa.int/About_Us/Careers_at_ESA/Young_Graduate_Trainees). The authors would like to acknowledge E. Teach, W. Kidd, A. Bonny and colleagues for inspiring the name of this project.



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




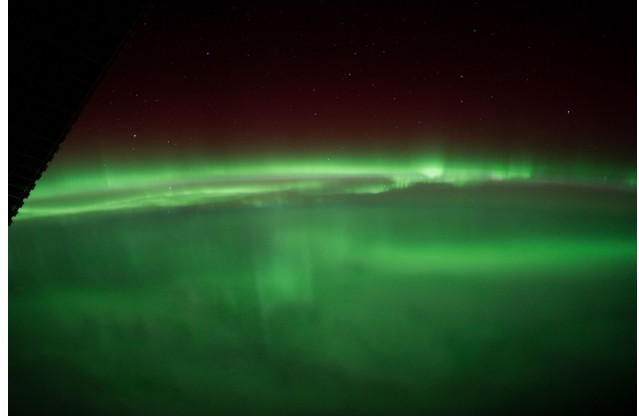

**Figure 1.** A typical astronaut image of the Aurora, image number ISS030-E-102282

Nishimura, Y., Lyons, L., Zou, S., Angelopoulos, V., and Mende, S.: Substorm triggering by new plasma intrusion: THEMIS all-sky imager observations, Journal of Geophysical Research: Space Physics, 115, doi:10.1029/2009JA015166, http://dx.doi.org/10.1029/2009JA015166, 2010b.

Obuchi, Y., Sakanoi, T., Yamazaki, A., Ino, T., Okano, S., Kasaba, Y., Hirahara, M., Kanai, Y., and Takeyama, N.: Initial observations of auroras by the multi-spectral auroral camera on board the Reimei satellite, Earth, Planets and Space, 60, 827–835, doi:10.1186/BF03352834, http://dx.doi.org/10.1186/BF03352834, 2008.

Paschmann, G., Haaland, S., and Treumann, R.: Chapter 4 - In Situ Measurements in the Auroral Plasma, Space Science Reviews, 103, 93–208, doi:10.1023/A:1023082700768, http://dx.doi.org/10.1023/A%3A1023082700768, 2002.

Skrutskie, M. F., Cutri, R. M., Stiening, R., Weinberg, M. D., Schneider, S., Carpenter, J. M., Beichman, C., Capps, R., Chester, T., Elias, J., Huchra, J., Liebert, J., Lonsdale, C., Monet, D. G., Price, S., Seitzer, P., Jarrett, T., Kirkpatrick, J. D., Gizis, J. E., Howard, E., Evans, T., Fowler, J., Fullmer, L., Hurt, R., Light, R., Kopan, E. L., Marsh, K. A., McCallon, H. L., Tam, R., Van Dyk, S., and Wheelock, S.: The Two Micron All Sky Survey (2MASS), Astronomical Journal, 131, 1163–1183, doi:10.1086/498708, 2006.

Stein, G. P.: Lens distortion calibration using point correspondences, in: Computer Vision and Pattern Recognition, 1997. Proceedings., 1997 IEEE Computer Society Conference on, pp. 602–608, IEEE, 1997.

Syrjäsuo, M., Pulkkinen, T., Janhunen, P., Viljanen, A., Pellinen, R., Kauristie, K., Opgenoorth, H., Wallman, S., Eglitis, P., Karlsson, P., et al.: Observations of substorm electrodynamics using the MIRACLE network, in: Substorms-4, vol. 238, p. 111, 1998.

Thébault, E., Finlay, C. C., Beggan, C. D., Alken, P., Aubert, J., Barrois, O., Bertrand, F., Bondar, T., Boness, A., Brocco, L., Canet, E., Chambodut, A., Chulliat, A., Coïsson, P., Civet, F., Du, A., Fournier, A., Fratter, I., Gillet, N., Hamilton, B., Hamoudi, M., Hulot, G., Jager, T., Korte, M., Kuang, W., Lalanne, X., Langlais, B., Léger, J.-M., Lesur, V., Lowes, F. J., Macmillan, S., Mandea, M., Manoj, C., Maus, S., Olsen, N., Petrov, V., Ridley, V., Rother, M., Sabaka, T. J., Saturnino, D., Schachtschneider, R., Sirol, O., Tangborn, A., Thomson, A., Tøffner Clausen, L., Vigneron, P., Wardinski, I., and Zvereva, T.: International Geomagnetic Reference Field: the 12th generation, Earth, Planets and Space, 67, 79, doi:10.1186/s40623-015-0228-9, http://www.earth-planets-space.com/content/67/1/79, 2015.


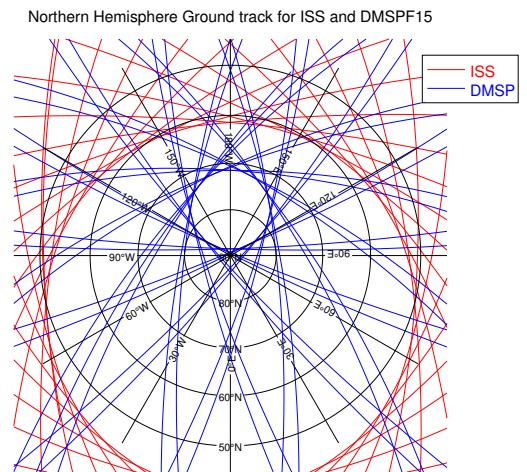

**Figure 2.** A comparison of the orbits of the International Space Station (red) and the DMSP F15 spacecraft (blue). The lower inclination of the ISS' orbit means it skims the auroral oval rather than cutting through it.

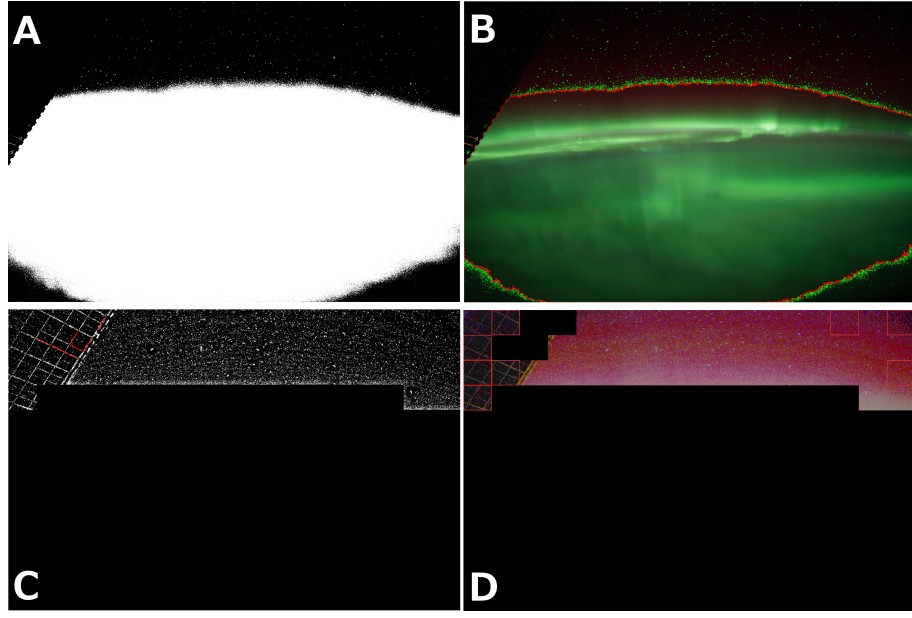

**Figure 3.** Stages of the automatic image masking process. A: The image is reduced to two colours to identify dark regions likely to be starfield. B: The resulting contour is applied to the original image. C & D: Image blocks not detected as starfield are masked and edge detection is applied to detect and mask linear features likely to be space station structure.



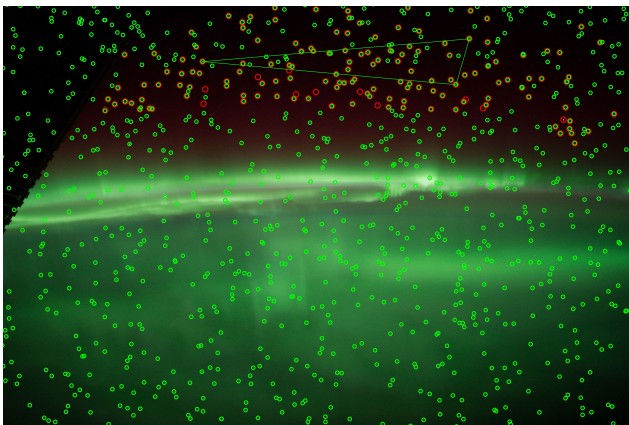

**Figure 4.** Results of applying astrometry.net to the image in fig. 1. Green circles represent the locations of catalogue stars, red circles stars that have been detected in the image and identified.

2012-01-25 09:28:44.060000 UTC

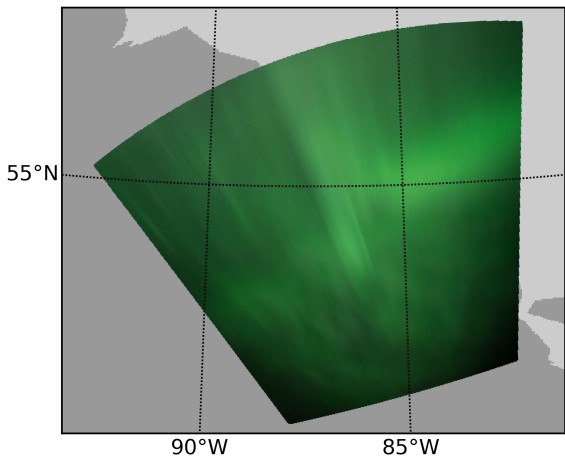

**Figure 5.** Results of georeferencing and projecting the image in fig 1 to 110km altitude.





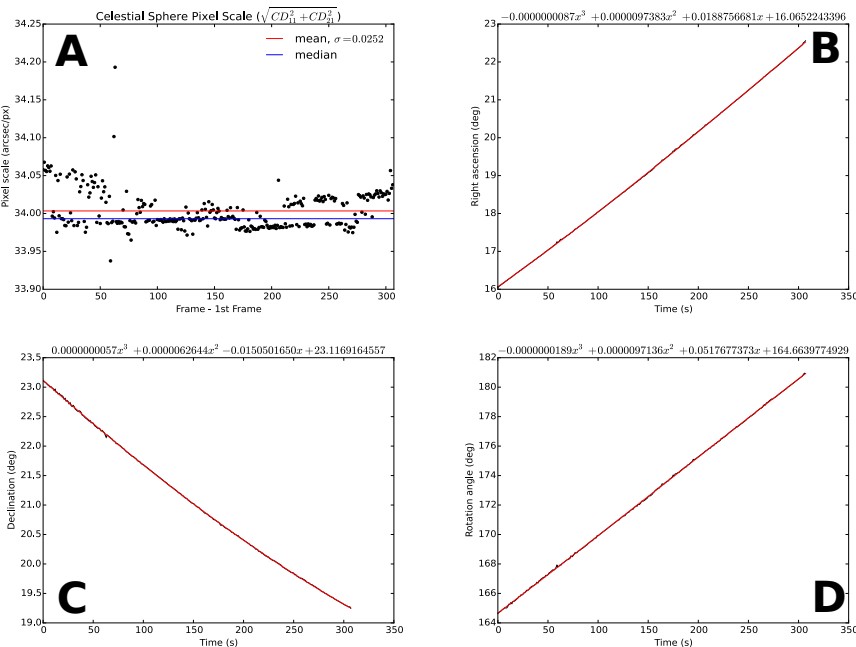

**Figure 6.** Astrometry parameters used for assessing the quality of a result. Figure 6A: The celestial sphere pixel scale for each image in a sequence (black) and its mean (red) and median (blue). Right ascension (6B), declination (6C) and rotation angle (6D) of the image centroid (black) and polynomial fits ot each of these (red).

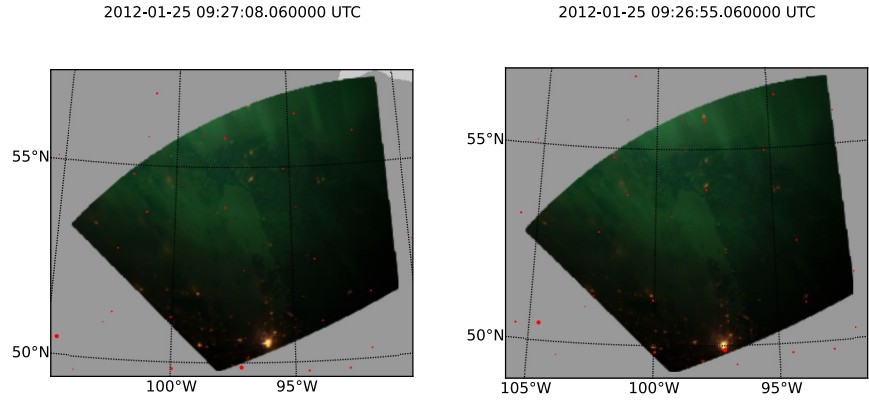

**Figure 7.** The left hand panel shows an image containing city lights projected to ground level rather than 110km, with the locations of cities overplotted in red. The right hand planel shows the same image when the timestamp has been corrected by 13 seconds to maximise agreement between the city lights and city locations.



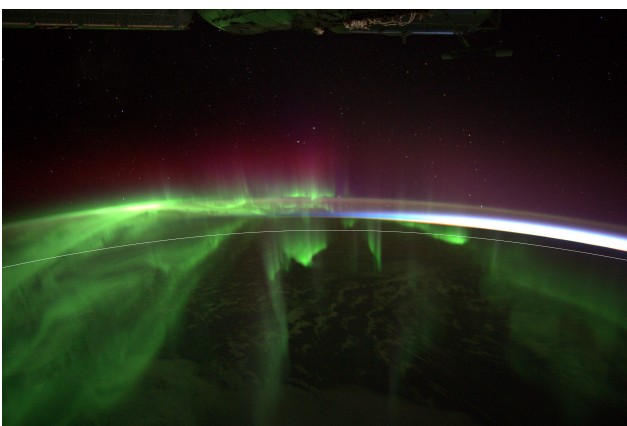

**Figure 8.** An image with the expected position of the Earth's horizon overplotted.

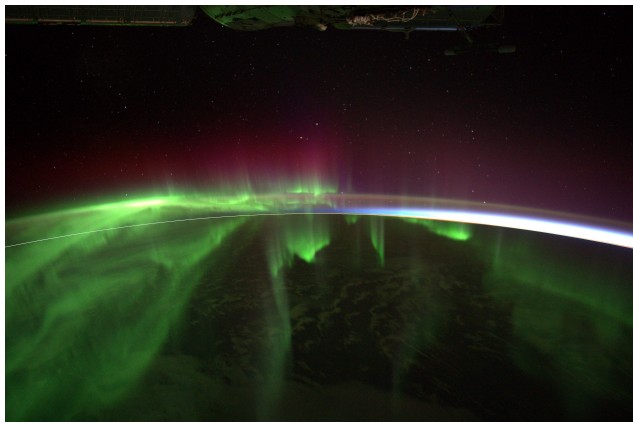

**Figure 9.** As fig.8 but with the timestamps corrected by 25 seconds. The horizon now matches its expected position much more closely.





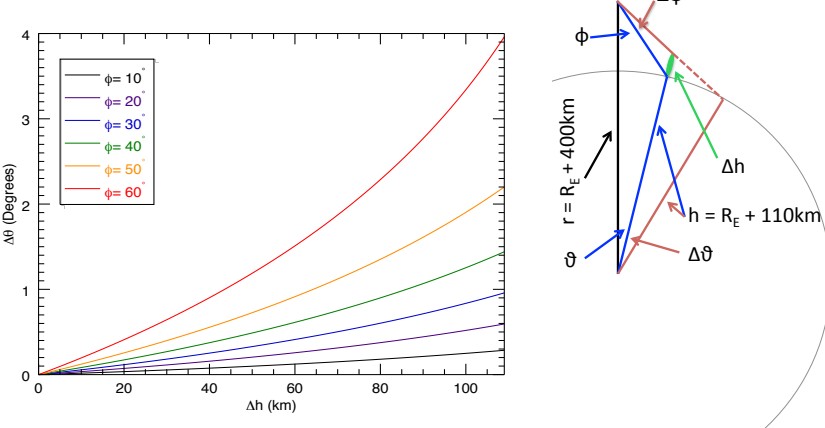

**Figure 10.** Each trace is the calulated angular extent of projected features ($\Delta\theta$) as a function of the altitude extent of the auroral emission ($\Delta h$) for a given value of the angle of line-of-sight away from nadir ($\phi$). The geometry is drawn to the right.

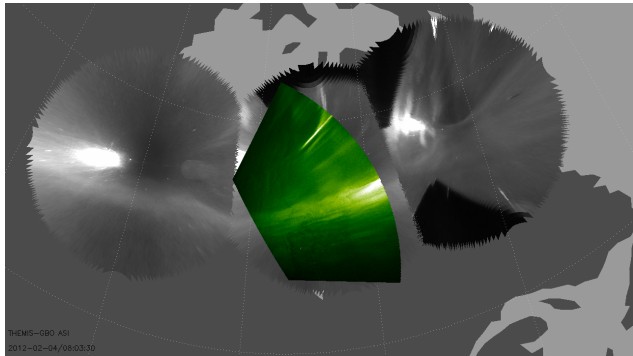

**Figure 11.** A comparison between a georeferenced ISS image (green) and data from the THEMIS ASI array (greyscale). The locations of the large scale features of the aurora are the same in the THEMIS and ISS data.

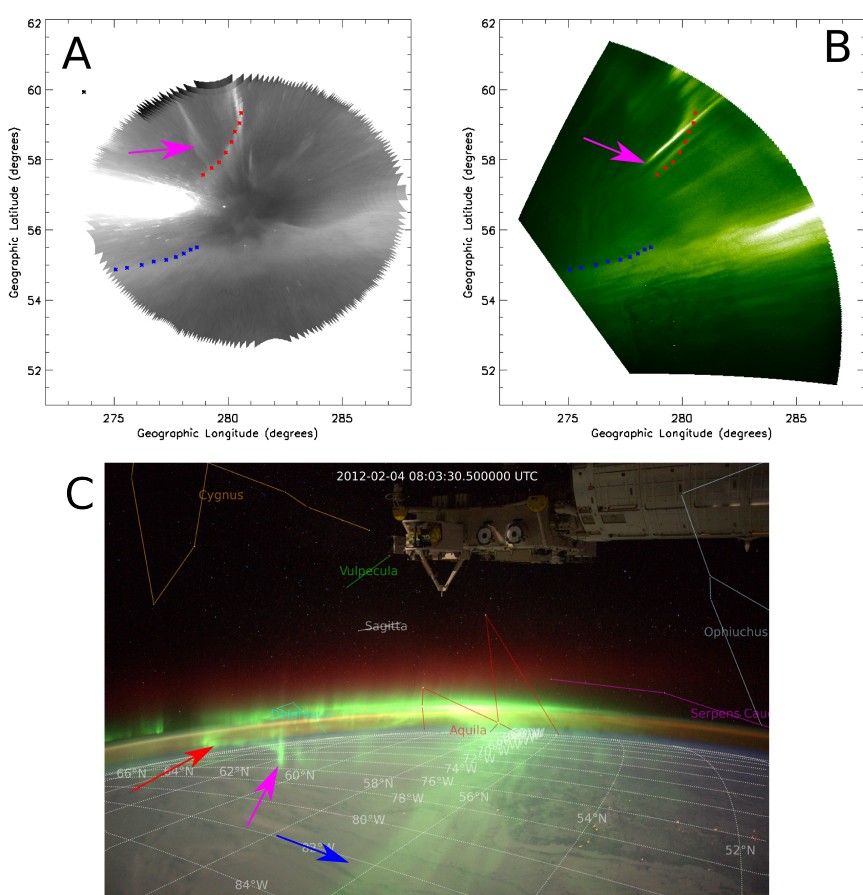

**Figure 12.** A more detailed comparison between the ISS image plotted in fig. 11 (Panel B) and the contemporaneous image acquired by the SNKQ THEMIS ASI (Panel A). The original ISS image is plotted in Panel C. Red and blue symbols trace the locations of the j shaped arc and northern edge of the main auroral arc, respectively, derived from their locations in the THEMIS image. The features are marked with the same coloured arrows in panel C. The magenta arrows point out a vertical feature projected very differently in A and B.



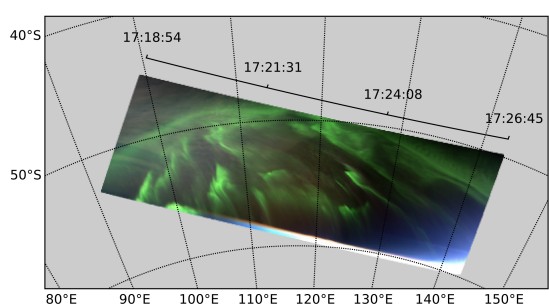

**Figure 13.** An example 'scanline' summary plot that is made available online for each image sequence.