# Peer review of "Automatic Georeferencing of Astronaut Auroral Photography"

_Geoscientific Instrumentation, Methods and Data Systems, 2015_

## Referee Comment (RC1) · Anonymous Referee #1 · 6 Apr 2016

Manuscript gi-2015-44 "Automatic Georeferencing of Astronaut Auroral Photography" by Riechert et al.

The authors report on a method for mapping auroral photographs taken from the International Space Station onto a geographic grid. The georeferenced images have been compared to scientific images from the ground-based network THEMIS. A lot of invaluable work has been put into this project to overcome issues of non-scientific imaging (lack of metadata, unknown pointing direction etc.) to make the photos scientifically useful. The work is carefully documented and I only have some rather
Specific comments:

- Adding some typical numbers based on already analyzed images would be very helpful. What is the typical size of a star in an image, in pixels? How much is it allowed to change across the image? How many stars are typically identified in an image and how many would be required as a minimum to produce an acceptable mapping?
- The images are ranked based on the pixel scale variation across the image into very good, excellent and bad (section 4.1). How many images out of the whole set of mapped images is included in each class? The same goes for the other errors. All error sources are very well described but a little note on how common they are in the analyzed set of data is missing.
- Maybe change "camera was did not move" to "camera did not move"
- The conclusion says that the comparison between ground-based and spacebased images can be within 12 arcminutes or better. This refers to the sample comparison (Figures 11 and 12) as the worst case scenario. Why is that? Have more than that one comparison with ground-based imager data been done? If yes, what was the overall performance?
- What is the triangle in Figure 4?
- Horizon lines in Figures 8 and 9 are a little hard to see, they could be thicker.
- The word calculated is misspelled in the caption of Figure 10. The geometry drawing is very good but the role of different colours could be described in the caption together with the meanings of the symbols (some of them are missing now).
- White blobs in the THEMIS images (Figures 11 and 12) are probably due to the Moon. That would be worth mentioning.
- The mapped ISS image in Figures 11 and 12 includes some almost vertical shadow-like features. Do you know where they come from?
- Is the vertical auroral feature marked by the magenta arrow in Figure 12 identified from the original image in panel C?

---

## Referee Comment (RC2) · Anonymous Referee #3 · 29 Apr 2016

The paper describes the methodology developed to make available georeferenced auroral images taken from the ISS. I found the paper to be well written. The material covered has value to complete direct aurora imaging from ground for studying the large scale behavior of the solar wind-magnetosphere-ionosphere system. It deserves publication.

---

## Author Comment (AC1) · 25 May 2016

We thank the referees for their careful reading of our manuscript and their positive comments. Below, we have addressed each of the specific suggestions made by Referee 1 (reproduced in quotation marks) in turn.

"Adding some typical numbers based on already analyzed images would be very helpful. What is the typical size of a star in an image, in pixels? How much is it allowed to change across the image? How many stars are typically identified in an image and how many would be required as a minimum to produce an acceptable mapping?"

The size of the stars in the images depends on a number of factors beyond the properties of the star it self. Camera sensor resolution and the lens used affect star pixel size, as does the exposure time (longer exposures lead to stars occupying more pixels because of ISS motion, and bloom, for example). IN a typical 12mega pixel image, bright stars occupy 10-12 px, for 16 mega pixel images this is more like 12-14, however since astrometry.net is resolution-independent this doesn't affect the use of the method.

In a distortion corrected image, star size doesn't change at all across the image, and even if the distortion correction is imperfect this doesn't really affect the size but rather the apparent location of a star.

The lowest number of stars that need to be identified in an image before the pointing can be successfully reconstructed is around 7, although the number s of stars actually identified can e several hundred, depending on the fraction of the image taken up with the Earth, ISS structure etc.

We've added this information to the manuscript

"The images are ranked based on the pixel scale variation across the image into very good, excellent and bad (section 4.1). How many images out of the whole set of mapped images is included in each class? The same goes for the other errors. All error sources are very well described but a little note on how common they are in the analyzed set of data is missing."

In fact the standard deviation of the celestial pixel scale we discuss is per image sequence rather than per image. It represents how stable the astrometric solutions are over a sequence - all else being equal the celestial sphere pixel scale (i.e. the plate scale of the image) should be the same in every image in a sequence, so any variations across a sequence represent inaccuracies in the astrometric solutions in one or more of the images. We've clarified this point in the manuscript.

57% of the sequences processed so far have standard deviations of < 0.1 arcseconds per pixel, 28% have standard deviations between 0.1. and 0.15 and only a single sequence has a standard deviation of greater than 0.5. We've also added this information to the manuscript.

Of the other sources of error, projection effects are always there to a certain extent, depending on the morphology of the aurora (the camera is never completely nadir-pointing), and there have been some timestamp inaccuracies in all of the analysed sequences apart from one, window distortion is also always there to a small extent, although this is represented by the standard deviation of celestial sphere the pixel scale. We've added a note with this information to the manuscript

"Maybe change "camera was did not move" to "camera did not move""

Thanks, updated.

"The conclusion says that the comparison between ground-based and space based images can be within 12 arcminutes or better. This refers to the sample comparison (Figures 11 and 12) as the worst case scenario. Why is that? Have more than that one comparison with ground-based imager data been done? If yes, what was the overall performance?"

This is somewhat clumsy phrasing on our part. We refer to the difference between the main auroral oval in Figure 12 (blue symbols) which is consistently mapped between the THEMIS and ISS images and the north-south aligned arc (red symbols) which has the 12 arcminute shift. Other images from the same sequence have similar characteristics. We haven't so far been able to find other sequences with easily identifiable, discrete auroral features while the ISS was overflying the THEMIS array.

We've updated the manuscript to clarify this.

"What is the triangle in Figure 4?"

The triangle represents one of the groups of stars whose relative positions astrometry.net uses to determine the plate scale and orientation of the image.

We've added this information to the caption of Figure 4

"Horizon lines in Figures 8 and 9 are a little hard to see, they could be thicker."

When using the horizon lines to correct the timing, the thinner they are the better the correction will be. However, we have added thicker lines to the figures for clarity.

"The word calculated is misspelled in the caption of Figure 10. The geometry drawing is very good but the role of different colours could be described in the caption together with the meanings of the symbols (some of them are missing now)."

These have been added.

"White blobs in the THEMIS images (Figures 11 and 12) are probably due to the Moon. That would be worth mentioning."

They are indeed saturation caused by the Moon. We've added this information to the caption of Figure 11.

"The mapped ISS image in Figures 11 and 12 includes some almost vertical shadow-like features. Do you know where they come from?"

The dark features at the top of the image (around 60 degrees latitude, 278-280 degrees longitude) are gaps in the aurora, other dark regions are most likely because of surface features. These appear dark because we are only plotting the green channel from the ISS image so as to highlight the auroral emission.

"Is the vertical auroral feature marked by the magenta arrow in Figure 12 identified from the original image in panel C?"

Yes it is. We've added this to the discussion of the figure.